# Regulation of Antimicrobial Effect of Hemicyanine-Based Photosensitizer via Supramolecular Assembly

**DOI:** 10.3390/nano12172905

**Published:** 2022-08-24

**Authors:** Huanxiang Yuan, Shaochuan Jia, Zelin Li, Jian Liu, Xiaoyu Wang, Ruilian Qi

**Affiliations:** 1Department of Chemistry, College of Chemistry and Materials Engineering, Beijing Technology and Business University, Beijing 100048, China; 2Institute of Chemistry, Chinese Academy of Sciences, Beijing 100090, China; 3School of Materials Science and Engineering, University of Science and Technology Beijing, Beijing 100083, China

**Keywords:** hemicyanine, photosensitizer, supramolecular assembly, photodynamic therapy

## Abstract

An intelligent “antimicrobial switch” has been constructed to reduce prolonged exposure of pathogenic bacteria to antibiotics, which could reversibly “turn off” or “turn on” the antimicrobial activity of hemicyanines through self-assembly or dis-assembly of cucurbit[7]uril (CB[7]). This assembly effectively inhibited the production of ROS under light, shielding the active site of hemicyanines and achieving on-demand antimicrobial ability. Moreover, CB[7] differentially inhibits ROS of molecules with different alkyl chain lengths, which provided reference for the subsequent design of materials with antimicrobial activity regulation, and could effectively delay or even prevent the development of pathogens resistance.

## 1. Introduction

A range of infections caused by drug-resistant pathogens have caused public health crises around the world due to the abuse/misuse of antibiotics [1,2]. The development of new antibiotics is one of the most effective ways to address this challenging problem, but traditional drug screening analyses and chemical synthesis strategies are resource intensive and lengthy, resulting in the development of new drugs always lagging behind the development of pathogenic bacterial resistance [3,4]. Therefore, more innovative strategies to modulate the antimicrobial activity of commercial antibiotics are urgently needed to slow or avoid the emergence of resistant bacteria.

In recent years, researchers have designed and developed a variety of supramolecular-based antibiotic switches that can precisely modulate the antimicrobial activity of drugs [5,6]. This approach does not require complex chemical modification of the antimicrobial molecule so it is the most simple and efficient antimicrobial regulation strategy compared with many other antibiotic design options [7,8], which requires firstly the design of supramolecular antimicrobial agents based on host-guest interactions, and secondly the control of the assembly/dis-assembly of the supramolecular system at the active spot to influence the interaction between its active site and pathogen in order to regulate the antimicrobial activity of the guest molecule [9,10,11].

Cucurbit[n]uril (CB[n]) is a cyclic compound with hydrophobic cavity, and the carbonyl groups at both ends easily form interaction sites with cations, which allows CB[n] to act as a supramolecular “host” of “guests” to binding metal ions or charged components of organic molecules through electrostatic interactions, hydrogen bonds, etc. [12,13,14,15,16]. Multiple commercially available drugs [17,18,19] or antimicrobial nanomaterials [20,21,22,23] can be encapsulated in CB[n] cavities to form highly stable inclusion complexes, which can not only reduce the non-specific toxicity of the materials, but also enhance or modulate its antimicrobial and antiviral activity on demand [17,24]. As a supramolecular drug different from other current extracellular drug mechanisms, it provides a simple, efficient and reliable idea for humanity to fight against bacteria, virus and tumor [25].

In this study, the stable hemicyanine supramolecular assembly between hemicyanines and CB[7] has been constructed and the principle of this antimicrobial switch to regulate antimicrobial activity was shown in Figure 1. Hemicyanines with different lengths of alkyl linkers (HC6 and HC10) that exhibit excellent photodynamic antimicrobial activity [26] were selected as guest molecules. The hydrophobic units of HC6/HC10 could insert into the cavity of CB[7] and the quaternary ammonium sites bind to the oxygen atom on the carbonyl of CB[7]. CB[7] could bury the positive charges and side chain alkyl groups of hemicyanines to limiting electrostatic interactions and hydrophobic interactions between hemicyanines and the surface of the pathogen. HC6 and HC10 exhibited different assembly behaviors, resulting in the diverse antimicrobial effects of the two assemblies. The photodynamic antimicrobial activity of HC6/CB[7] assembly dramatically reduce due to the shielding of interaction sites towards pathogens and ROS generation of HC6. When adding the competitor amantadine hydrochloride (AD), a more stable AD/CB[7] was formed to release HC6 and restored its antimicrobial effect. Therefore, this HC6/CB[7] could reversibly “turn off” or “turn on” the antimicrobial activity of hemicyanines through self-assembly or dis-assembly, which is an “antimicrobial switch” that can circularly regulate the antimicrobial effect. This strategy is simple, efficient and does not require any chemical modification of the active site of the antimicrobial agent. Not only can the goal of precise regulation of antimicrobial activity be achieved, but also it is expected to delay or even prevent the development of drug resistance in pathogenic bacteria, which is of great significance to reduce the accumulation of antibiotics in the environment.

## 2. Materials and Methods

### 2.1. Construction of Supramolecular Conjugated Complexes of HCs/CB[7]

Hemicyanine HC6 or HC10 were prepared in DMSO at 10 mM as a stock solution, while CB[7] was prepared in deuteroxide at 10 mM as a stock solution. Firstly, different volumes (5, 10, 15, 20, 30, 40, 50 μL) of CB[7] stock solution were added to deuteroxide to obtain homogenized solutions with the same volume of 495 μL by vortex oscillation. Then 5 μL of hemicyanine HC6 or HC10 stock solution was added to the above solutions, respectively, and supramolecular complexes with different CB[7] ratios were obtained by rapid vortex oscillation for 3 min (HCs:CB[7] = 1:1, 1:2, 1:3, 1:4, 1:6, 1:8, 1:10). A sudden change in colour can be seen during this process, with the colour gradually becoming lighter to colourless as the proportion of CB[7] increases.

### 2.2. Characterization of HCs/CB[7]

1 mL of HCs or HCs/CB[7] solution ([HCs] = 0.5 μM) was added to the quartz cell (1 cm × 1 cm) for particle size measurement and equilibrated for 2 min at 25 °C. The particle size of HCs/CB[7] was obtained by averaging three times in parallel. UV-Vis absorption spectra of HCs/CB[7] ([HCs] = 10 μM) solutions with different CB[7] ratios were tested using a JASCO V-550 spectrometer. The scanning speed was medium and the UV-Vis absorption spectra were tested from 200 nm to 800 nm. A fluorescence spectrophotometer was used to test the fluorescence emission spectra of HCs/CB[7] ([HCs] = 10 μM) solutions with different CB[7] ratios. 455 nm and 460 nm were chosen as the excitation wavelengths of HC6/CB[7] and HC10/CB[7], respectively. The scanning voltage was 700 V and the fluorescence emission spectra from 475 to 700 nm were tested. The obtained complexes of HCs/CB[7] in Section 2.1 were characterized by ^1^H NMR (Bruker Avance III 400 HD, Zurich, Switzerland).

### 2.3. Reactive Oxygen Species (ROS) Measurements

2,7-dichlorofluorescin diacetate (DCFH-DA, 10 mM) stock solution was obtained by dissolving 4.87 mg of DCFH-DA in 1 mL of ethanol. 50 μL 10 mM DCFH-DA stock solution was added with 450 μL ethanol and 2 mL NaOH aqueous solution (0.01 M) and then activated at room temperature for 30 min protecting from light. 10 mL of 1 × PBS (10 mM, pH = 7.4) was added to obtain a final concentration of 40 μM 2,7-dichlorodihydrofluorescein (DCFH) solution on ice and was kept away from light. Since DCFH is oxidized to 2,7-dichlorofluorescein (DCF, excitation peak at 488 nm, emission peak at 524 nm, quantum yield 90%) with high quantum yield when it encounters ROS, the ability of the material to sensitize the surrounding oxygen to produce ROS can be determined by testing the fluorescence emission intensity at 524 nm, and the change in DCF fluorescence intensity and ROS production are positively correlated. Activated DCFH solution and different ratios of HCs/CB[7] (final concentration of 0.5 μM) were added to a disposable fluorescent plastic cuvette, and then irradiated under white light (5 mW/cm^2^) for 5 min, and the fluorescence emission spectra of DCF at 500~700 nm under 488 nm excitation were recorded every minute. In the blank group, the HCs/CB[7] was replaced with the same volume of ultrapure water and the rest of the test conditions were unchanged.

### 2.4. Regulation of Antibacterial Effect

The monoclonal colonies from the solid medium were transferred to the 10 mL liquid medium adapted to the different species of microbes (*E. coli* with ampicillin resistance require an additional 10 μL of 50 μg/mL ampicillin) and incubated for 8~12 h at a constant temperature (37 °C for bacteria, 30 °C for fungi) in an oscillator with 180 rpm. 2 mL of microbial solution was centrifuged and the precipitate was washed twice with 1 × PBS (10 mM, pH = 7.4). The remaining precipitate was resuspended in 1 × PBS and diluted to an optical density of 1.0 at 600 nm (OD_600_ = 1.0) for bacteria and 1.5 at 600 nm (OD_600_ = 1.5) for fungi.

100 μL pathogen suspension was mixed with HCs or HCs/CB[7] in the dark, respectively. The volume was replenished to 500 μL with 1 × PBS and incubated for 30 min at the appropriate temperature (37 °C for bacteria and 30 °C for fungi). After the incubation, the dark group was placed in the dark and the light group was exposed to white light (65 mW/cm^2^) for 10 min. All experimental groups were diluted with 1 × PBS and 100 μL was spread on 90 mm solid medium, then the colony forming units (CFU) were recorded after culturing for 18~20 h. The HC6/CB[7] + AD group was added with the AD solution to disassemble HC6/CB[7] (HC6:CB[7]:AD = 1:10:50) after 20 min of interaction with *E. coli*. Then the dark group was continued to incubate for 10 min protected from light, while the light group was irradiated for 10 min under white light (65 mW/cm^2^) followed by the same dilution and coating treatment. The control group was the pathogen sample which only interacts with AD solution, and the blank group was the pathogen sample without any treatment. The antimicrobial properties (inhibition ratio, *IR*) of HCs/CB[7] were calculated according to the following equation.
$$IR=\frac{C_{0}-C}{C_{0}}\times 100\%$$
where *C* is the colony-forming units (CFU) of the experimental groups treated with AD alone, HCs, HCs/CB[7] or with HCs/CB[7] in the presence of AD, and *C*_0_ is the CFU of the blank group.

### 2.5. Measurement of Zeta Potential

*E. coli*, *S. aureus* or *C. albicans* were incubated with HCs or their corresponding supramolecular complex HCs/CB[7] in 1 × PBS (10 mM, pH = 7.4) for 30 min at appropriate temperature (37 °C for bacteria and 30 °C for fungi), respectively. The HCs or HCs/CB[7] that did not interact with the pathogen were removed by centrifugation (7100 rpm, 3 min), and the precipitate of the pathogen was resuspended in 1 mL of ultrapure water and placed on ice. In the control experiment, pathogens without HCs or HCs/CB[7] were treated with 1 × PBS (10 mM, pH = 7.4) alone in the same process with that of experimental groups.

### 2.6. Measurement of Fluorescence Microscopy

The procedure in the zeta potential measurement was repeated to obtain a precipitate of *E. coli* after interaction with different complexes of HCs/CB[7], then 10 μL of 1 × PBS (10 mM, pH = 7.4) was added to resuspend the precipitate to obtain various bacterial suspensions. 5 μL of them was dropped onto a slide, covered with a coverslip and placed under a 100× fluorescent microscope for observation.

## 3. Results and Discussion

Firstly, ^1^H NMR was applied to characterize the supramolecular non-covalent bonding process between these two cationic antimicrobial fungicides and CB[7]. As shown in Appendix A, all of the proton signals of CB[7] (5.84~5.88 ppm, 5.58 ppm, 4.25~4.30 ppm) could be evidently detected after CB[7] was added to the aqueous solution of hemicyanine derivatives. The assembly of HC6/HC10 and CB[7] reached saturation at the molar ratio of 1:4 as exhibited in the unchanged positions of proton peaks. The direction and magnitude of the shifts provides confirmation of the proposed geometries of these complexes in accord with the well-known shielding region inside the CB[7] cavity and the desheilding region nearby the portals [27]. Take the assembly of HC6 and CB[7] as an example. Specifically, the proton signals (c) of phenyl nearby the vinyl have an upfield shift, indicating this part of phenyl unit is inside the CB[7] cavity, and the chemical shifts of the vinyl protons (d) almost remain unchanged, implying one of the protons is inside the CB[7] cavity and the other is nearby the portal of CB[7]. While the proton signals of pyridinium (a, b), dimethyl amino (g) and phenyl part (e) close to the amino group have a downfield shift, demonstrating these protons are nearby the portals of CB[7]. Therefore, the CB[7] molecule is in the middle of phenyl and vinyl units. This result is consistent with the previous report [27]. Furthermore, the chemical shifts of the protons (h, i) on the alkyl have an upfield shift, indicating they are all inside the CB[7] cavity. The above observations preliminarily proved the formation of HCs/CB[7] complex, with CB[7] possibly wrap around hemicyanine derivatives. Because of the positive charges and hydrophobic structure, HCs alone could form aggregates in aqueous solution to obtain a measurable diameter by DLS. The particle size distribution diagram could demonstrate the size change in hemicyanine molecules in aqueous solution upon combining with CB[7] (Figure 1a,d and Appendix A). The average diameters of HC6 and HC10 alone is about 37 nm and 46 nm, while the encapsulation effect of CB[7] on hemicyanine molecules significantly increases the average diameters of HC6/CB[7] and HC10/CB[7] complex to 168 nm and 177 nm, which further corroborate the interactions between HCs and CB[7].

The UV-Vis absorption and fluorescence emission spectra of the supramolecular assemblies based hemicyanine derivatives were tested after adding different proportions of CB[7] ([hemicyanine]:[CB[7]] = 1:1, 1:2, 1:3, 1:4, 1:6, 1:8 and 1:10) to better understand the interactions between them. With the increase in CB[7] concentration, the maximum absorption wavelength of hemicyanine derivatives gradually indicated a blue-shift about 100 nm, and the colour of the aqueous solution changed from orange to colourless by degrees (Figure 1b,e). Moreover, the fluorescence intensity of HCs/CB[7] complex was higher than that of hemicyanine alone, but the fluorescence emission intensity first increased and then decreased with the increase in CB[7] concentration without the variation in spectral shape. Specifically, the fluorescence intensity reached the maximum when HC6:CB[7] = 1:3 and HC10:CB[7] = 1:4, possibly because the assembly of HC6/HC10 and CB[7] was saturated at these ratios in the consistence with the NMR results (Figure 1c,f). The decrease in fluorescence intensity after assembly could ascribe to the absorption blue-shift of HC/CB[7] compared to HC. Therefore, the variation in fluorescence intensity suggested that the supramolecular self-assembly process changed the original spatial conformation of hemicyanine molecules, and the appropriate proportion of CB[7] could reduce the aggregation degree of hemicyanines to significantly lowering its fluorescence self-quenching ability. All the above results revealed that supramolecular assemblies could be successfully prepared by simple mixing of hemicyanines and CB[7]. The whole process took only 3 min without other fussy experimental equipment. This supramolecular assemblies have completely different spatial conformation and aggregation states from hemicyanine derivatives, which possess the potential to vary the form and manner of its interaction with pathogenic bacteria to achieve the regulation of antimicrobial effect.

Since the effect of photodynamic antimicrobial therapy (PDT) is closely related to the production capacity of reactive oxygen species (ROS) of photosensitizers, the production efficiency of ROS sensitized by supramolecular complex of HCs/CB[7] was tested. 2,7-dichlorofluorescein diacetate (DCFH-DA) was selected as the probe for detecting the total ROS. When DCFH-DA was treated with NaOH, it could be hydrolyzed into 2,7-dichlorodihydrofluorescein (DCFH), which could be oxidized by ROS to form fluorescent 2,7-dichlorofluorescein (DCF). The ROS production capacity of the photosensitizer was revealed by the elevation of fluorescence signal at 524 nm. As displayed in Figure 2, the photosensitization ability of HC6 decreased with the increase in CB[7] concentration. The HC6/CB[7] complex (HC6:CB[7] = 1:10) almost did not produce ROS, which demonstrated that CB[7] could inhibit the ROS production effect of HC6. On the contrary, the photosensitization efficiency of HC10/CB[7] in different proportions varies irregularly and fluctuates within a certain range without remarkable decline, indicating that CB[7] hardly changes the ROS generation of HC10 itself. It could be attributed to the relative long alkyl linker containing 10 carbon atoms of HC10 in which CB[7] was moving in a dynamic process for the ineffective shielding of ROS generation. These results proved that the length of alkyl chains would affect the ROS regulation ability of the assemblies, which offered assistance for the subsequent design of supramolecular assembly materials with the adjustment ability of ROS.

In order to explore the regulation ability of supramolecular assemblies HCs/CB[7] on antimicrobial effect, Amp^r^ *E. coli* (TOP 10), *S. aureus* and *C. albicans* were selected as representative pathogens of gram-negative bacteria, gram-positive bacteria and fungi, respectively. The zeta potential changes of different microbes before and after binding hemicyanines or their supramolecular assemblies were studied. As shown in Figure 3, Appendix A, pathogens all had varying amounts of negative charges on their surfaces, and fungi was less than that of bacteria. When the pathogens interacted with the positively charged hemicyanines, respectively, some of the negative charges on the surface of them were masked while the positive charges of the hemicyanines were exposed, so the zeta potential indicated a positive shift. Due to the presence of a denser and thicker cell wall in *C. albicans*, the positive charges of the hemicyanines were easily encapsulated resulting in insignificant changes in zeta potential. In addition, as the alkyl chain of HC10 was longer than that of HC6, which positive charges were more easily exposed after inserting into pathogens to showing a more positive potential [28]. When HCs/CB[7] complex bound to pathogenic bacteria, the positive charges of hemicyanines were shielded by CB[7] to limiting the positive shift of zeta potential of microbes. This phenomenon suggested that the positive charges and alkyl chains of the hemicyanines could be effectively encapsulated by CB[7], preventing cationic groups from being inserted into the cell membrane of pathogens, which prevented the combination of the two through electrostatic and hydrophobic interactions, leading to the recovery of the surface potential of the pathogens. This assembly strategy could avoid the long-term exposure of pathogens to antimicrobial materials for the further non-proneness to the development of drug-resistance.

Subsequently, the shielding potentiality of CB[7] on the photodynamic antimicrobial activity of hemicyanine derivatives towards the three representative pathogen species was investigated by plate antimicrobial counting experiments to analyze the differences in the antimicrobial activity of hemicyanines or HCs/CB[7]. The number of colonies on agar plates provided visual evidence (Appendix A). The dark toxicity of the supramolecular assemblies was weak and independent of the concentration of CB[7]. Whereas upon light irradiation, the colonies on the plates gradually increased with the increase in CB[7] concentration in the HC6/CB[7]. But the assembly of HC10 containing longer alkyl linker with CB[7] did not influence the growth of microbes which displayed almost as many colonies as the control group of HC10 alone in both dark and light conditions (Appendix A). Figure 4 shows a histogram of the inhibition rates of hemicyanines and their assemblies with various proportional CB[7], where various concentrations of CB[7] barely changed the weak dark toxicity of hemicyanine molecules under dark condition. The antimicrobial effect of the hemicyanines gradually diminished with the increase in CB[7] concentration under 65 mW/cm^2^ white light irradiation, and when the ratio of HC6:CB[7] was 1:10, the photodynamic antimicrobial effect of HC6 was completely blocked with low killing efficiencies of 9.6%, 1.9% and 2.5% against *E. coli*, *S. aureus* and *C. albicans*, respectively, which means that the antimicrobial activity of the hemicyanine derivatives was “switched off” by this super-intelligent antimicrobial assembly. This shielding effect on the antimicrobial ability was mainly due to the encapsulation of CB[7] which not only weakened the ability of the HC6 to bind with pathogens, but also hindered the contact of HC6 with oxygen and the diffusion of ROS resulting in the demonstration of reduced ROS generation ability and a significant decrease in antimicrobial rate. Furthermore, as shown in Figure 4a–c, the trend of decreasing inhibition of pathogens with increasing concentrations of CB[7] was consistent with that of shielding ROS production capacity of HC6. For HC10/CB[7] (Figure 4d–f), although the encapsulation of CB[7] reduced the binding of C10 to pathogenic bacteria, it did not affect the ability of HC10 to sensitize oxygen to produce ROS leading to almost unchanged photodynamic antimicrobial effect of HC10. The results of the antimicrobial experiments above demonstrated that the regulation of ROS production capacity played a dominant role in the shielding of the antimicrobial effect of the supramolecular assemblies. Moreover, the assembly between different alkyl linker lengths of hemicyanines and CB[7] contributes to differences and modulation in disinfection activity and only appropriate length of alkyl linker was positive for the assembly regulation, which provides ideas for designing antimicrobial strategies with reactive oxygen species regulation ability.

To better visualize the combination of the supramolecular assemblies to the pathogens, fluorescence microscopy photographs of the *E. coli* were taken before and after adding into HCs/CB[7] in the absence and presence of light irradiation. As shown in Figure 5, the fluorescence changes of *E. coli* in the light and dark groups stained with HCs or HCs/CB[7] were observed in the fluorescent field. As the concentration of CB[7] increased, more CB[7] could be snapped onto HC6/HC10 to encapsulate their alkyl chains and positive charges, leading to weakened binding with pathogens. Although the fluorescence intensity of HCs/CB[7] is stronger than that of HCs alone, the HCs/CB[7] that unbound on the microbes was washed before fluorescent imaging. Therefore, fluorescence intensity was significantly reduced in both dark and light conditions. Interestingly, the fluorescence of bacteria that interacted with both HC6/CB[7] and HC10/CB[7] (Appendix A) in the light group was significantly stronger than that in the dark group when CB[7] concentration was low. This phenomenon may be due to that the positive charge of hemicyanines could not be completely shielded, and they could still combine with microorganism with simultaneous ROS production under light irradiation to cause the death of pathogenic bacteria which promoted HCs to stain the nuclei of microorganisms to exhibit brighter fluorescence [29]. The results of this experimental phenomenon were consistent with antimicrobial experiment, which once again demonstrated the shielding mechanism of supramolecular assemblies.

Since HC6/CB[7] has the ability to shield the photodynamic antimicrobial activity of HC6, this complex was selected for subsequent dis-assembly experiments. AD that had tight association with CB[7] was utilized to dis-assemble HC6/CB[7] to release the positive site and ROS generation capacity. Both absorption and emission spectra demonstrated that HC6/CB[7] was unassembled after the addition of AD (Figure 6a,b). Specifically, the maximum absorption wavelength of HC6/CB[7] + AD was back to 455 nm which was the characteristic absorption peak of HC6, and the fluorescence emission intensity of HC6/CB[7] + AD reduced close to that of the initial HC6. To evaluate the recovery of antimicrobial activity of HC6 after adding AD into HC6/CB[7] complex, antimicrobial experiment against *E. coli* was performed. Under the white light irradiation (65 mW/cm^2^), HC6 showed extremely high bactericidal efficiency against *E. coli*, while for HC6/CB[7] less than 10% inhibition was achieved. The competitive molecule AD was added to bacteria treated with HC6/CB[7] to conduct in situ disassembly for the release of antimicrobial ability. The inhibition ratio was then restored to approximately 70% after disassembly of HC6/CB[7] (Figure 6d), which was more visually evidenced by the number of colonies on LB plates (Figure 6e). Meanwhile, Figure 6c illustrated that bacterial growth was virtually unaffected by the experimental concentration of AD. Scanning electron microscopy characterization (Appendix A) demonstrates the morphology change in *E. coli* treated with HC6 and HC6/CB[7] in the absence and presence of light, which indicates that the antimicrobial effects of HC6 and HC6/CB[7] rely on the damage of microbes. These results suggested that the antimicrobial activity of the cationic conjugate molecule HC6 could be reversibly “switched off” and “switched on” by simple supramolecular host-guest interactions.

## 4. Conclusions

In this study, the simple “antimicrobial switch” based on the principle of supramolecular self-assembly was successfully constructed, which achieved the regulation of photodynamic antimicrobial activity of HC6 through the reversible “assembly” and “disassembly” process between organic conjugated hemicyanines and large ring molecule CB[7], and it was also found that the regulation of ROS production capacity played a dominant role in the regulation of antimicrobial activity. In addition, an appropriate proportion of CB[7] could well bury the alkyl chains and positive charges of the HCs to weakening their binding with pathogens through hydrophobic and electrostatic interactions. More importantly, molecules of different lengths could dynamically regulate the antimicrobial properties of the assembled complex. This supramolecular assembly strategy did not require any chemical modification of the active site of existing antimicrobial molecules and only 3 min was needed to achieve the goal of precise modulation of antimicrobial activity. Overall, this supramolecular strategy provided an idea to regulate active sites on demand, effectively avoiding long-term exposure of antimicrobial materials and availably prevent the emergence of drug-resistant bacteria.

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
