# Peer review of "Regulation of Antimicrobial Effect of Hemicyanine-Based Photosensitizer via Supramolecular Assembly"

_nanomaterials, 2022, doi:10.3390/nano12172905_

Round 1
Reviewer 1 Report
The manuscript by Qi and colleagues is an interesting piece of work that describes the effects of a supramolecular approach on the activity as photosensitizers of two derivatives of Hemicyanine. By using a cucurbit[7]uril as host they demonstrate the possibility of switching off the biological activity against three different types of pathogens as consequence of the decrease of ROS production upon inclusion of the dye. Moreover, they show the possibility of restoring the antimicrobial activity by adding an adamantane derivative that due to its higher affintiy for CB[7] displaces the HC. In this way, this can perform its function of photosensitizer killing the patogenes by irradiation.
The content of the paper is certainly of interest for the readership of Nanomaterials journal, but some revisions are necessary to make it suitable for publication.
In general, a deep revision of the English language is necessary. Several sentences are grammatically incorrect and in some cases (for example at lines 47-49, 128-129, 255-256) difficult to understand.
Going to specific points:
- The section of Materials and Methods is difficult to read. The followed procedures are described in a not sufficiently clear way, in particular that relative to the regulation of the antibacterial effect. For example, but not exclusively, at line 124, it is written “after removal…” but it is not clear what was removed; at line 128 is written “The unassembled group with the addition of competing molecules was added to the AD solution”, but it is not clear what are the competing molecules.
- The part relative to the NMR studies of the host-guest complexation presents some problems. A minor one is that in SI the solvents of the samples must be explicitly mentioned in the figure captions. More important, with respect to the description given at line 160, the aromatic and vinyl protons of HC are, actually, downfield shifted, and the protons of the aliphatic chains move, even if not regularly in all cases, to lower fields.
As control, the effect of the addition of D2O should be monitored to rule out that the change in the signal chemical shifts is due to it or partially to it or to chages in pKa of anilinium.
In general, on the basis of previously reported data (NewJ.Chem., 2014, 38, 3600), it is not clear how the behavior of the signal chemical shift explains in this submitted work the formation of the host-guest species and its geometry.
- The size of the particles formed by HC alone detected by DLS indicates that a self-assembly process takes place. It is important to specify this phenomenon and also explain if the increased size of the samples containing the HC/CB[7] is due to “single” host-guest species or to an assembly of several units of it.
- At line 182 the authors write: “The decrease in fluorescence intensity after assembly saturation might ascribe to the furtherance in the solubility of hemicyanine molecules by CB[7], resulting in non-radiative energy dissipation”. But at a 1:10 HC/CB[7], is it not expected that all the HC is complexed? Why would there be a furtherance of solubility of HC?
- At line 327, the sentence is not clear to me, because previously it was shown that the inclusion of HC in CB[7] switches on its fluorescence, then I would expect higher fluorescence intensity in this experiments increasing the HC/CB[7] concentration. I understand the issue of the lower adhesion of HC/CB[7] to microorganism surface, but the general fluorescence should be higher by adding CB[7]. Perhaps is there something done in the experiments that should be described to explain better the behavior observed?
- In figure 6, in the circular scheme of switching-on and –off, it could be useful to report over the arrows what is necessary to make possible the different steps (CB[7] to switch-off, AD to switch-on and the
- Two minor points: in Scheme 1, as for CB and AD, it would be useful to schematically represent the inclusion complexes; in Figure 3 caption, the reported zeta potential diagrams are actually relative to the microorganisms in absence and in presence of HC6, HC10 and their host-guest complexes and not vice versa as currently written.
Reviewer 2 Report
Version:1.0 StartHTML:0000000165 EndHTML:0000039767 StartFragment:0000026599 EndFragment:0000039727 SourceURL:file:///Y:\biralat\nanomaterials_202208\kmd.docx
The authors present two intriguing hemicyanin-cucurbit[7]uril (CB[7]) systems. It is shown that the photosensitizer capacity as well as the membrane-binding capacity of hemicyanins can be limited by complex formation with the macrocycle. Since complex formation can simply be initiated by the addition of CB[7] and then reversed by supplying a competitive inhibitor, amantadine hydrochloride (AD), the system can be switched on an off at will. This is a nice finding, although the context, methods and in some case even the wording is strongly reminiscent of the works of Prof. Shu Wang (see for example https://onlinelibrary.wiley.com/doi/full/10.1002/anie.201504566) - but the present paper focuses on the antimicrobial effect exerted through generation of reactive oxygen species and a distinct hemicyanin target, thus it can be considered as an independent study.
My comments and questions are the following:
· line 39-40: Instead of “the carbonyl groups at both ends are easy to form interaction sites with cations” write something like: the carbonyl groups at both ends easily form interactions with cations…
· line 40-41: Instead of “which allows CB[n] to act as a supramolecular "host" of "guests" to binding metal ions or charged components of organic molecules through hydrophobic interaction of cavities”, maybe: which allows CB[n] to act as a supramolecular "host" of "guests" to binding metal ions or charged components of organic molecules through electrostatic interactions, hydrogen bonds etc.
· line 43: commercially available drugs
· line 46: In Refs 24 and 25 not a single word concerning the antimicrobial or antiviral effect of CB[n]-s can be found – better citations should be found to support this sentence or the sentence should be rephrased
· line 48: something is missing… perhaps: “…humanity to fight against ….
· line 51: As I understood, hemicyanines are not antimicrobial agents themselves, they are photosensitizers of a photodynamic antimicrobial process. This should be clarified from the beginning, or if indeed a self-standing antimicrobial/antiviral capacity of these molecules is known then that should by supported by appropriate references.
· Scheme 1. I was not convinced that the phenyl and ethylene segments of HC6/HC10 that are indicated by the light-blue ovals can be singled out as binding sites of CB[7]. Based on crystal structures of protein-CB[7] complexes (see Protein Data Bank structures: 6su0, 7p2j, 6f7x, 3q6e, for example) I would guess that the terminal dimethyl amine can also dock into the heterocycle, for example.
· In connection with the previous point, the NMR analysis of the complexes should be presented in more detail. I suggest replacing Scheme 1b which (in the absence of assignation or ppm scale) is quite baffling, with Figure S1 and S2 – maybe these could form a new Figure 1. The spectra should be analyzed in greater detail. Assignment of peaks within the complexes should be attempted too, as well as those of the broadening, shifting, emerging and disappearing peaks. In support of my previous argument, it can be seen – for example, that the signal of the hydrogens of the N,N-dimethyl-termini (marked as “g”) shift considerably too. Also, is there any evidence that the CB[7] molecules “wrap” around the hemicyanine derivatives (line 163) instead of just forming various interactions with them (especially with the middle segment)?
· Description of the NMR experiments is missing from the Methods section – it should be added.
· line 77-78: I am unfamiliar with the expression “reserve liquid”, maybe others are too (maybe stock solution?). Please add some clarification.
· line 100-101: instead of using “Add” and “keep away from light” it would be more in line with the rest of the text if you wrote: …was added… and “kept away from light”.
· line 106-107: The entire sentence “The amount of ROS production was positively correlated.” is already in the previous sentence. This 2nd occurrence should be removed.
· Based on the UV-vis spectra (Figure 1b, 1e) the transfer to a stable assembly becomes complete at HC6:4CB[7] ratio (peak near 450nm disappears, peak near 325nm appears) and stays stable as the CB[7] content increases further. This is in slight disagreement with the interpretation of the fluorescence emission spectra (Figure 1c, 1f + lines 182-187). Could this be resolved somehow?
· Figure 5 is very hard to see. In the figure caption, you could add some explanation as to what it is that we should see.
Reviewer 3 Report
In the manuscript "Regulation of Antimicrobial Effect of Hemicyanine-based photosensitizer via Supramolecular Assembly", the authors report the assembly of hemicyanines (with C6 or C10 alkyl chain) with cucurbit[7]uril (CB[7]), allowing to tune the generation of ROS. Overall this supramolecular strategy is promising for on-demand antimicrobial properties, which could avoid long-term exposure of antimicrobial materials leading to drug-resistant bacteria. This work is original. Nevertheless, there are some issues that should be addressed before publication.
As a consequence, I recommend to accept this manuscript for publication in Nanomaterials after major revision.
1- Characterization by DLS is appropriate for spherical particles. It would be relevant to perform transmission electron microscopy of the complex between hemicyanines and CB[7] to (i) have information about the morphology of the self-assembled structures, (ii) confirm the size assessed by DLS, and (iii) know if DLS is an appropriate technique to determine the size of the complex.
2- Figure 1c, 1f, sentence “The decrease…energy dissipation”: in my opinion, the decrease of fluorescence intensity is due to a change of the absorbance properties after complexation with CB[7], as seen in Fig. 1b, 1e.
Figure 1b, 1e: it would be interesting to show the UV-Vis spectrum of CB[7] for comparison. Same remark for the emission spectrum in Fig. 1c, 1f.
Fig. 1b: why is the absorbance of the 1:1 complex higher than that of 1:0?
3- What is the stability of the complex upon light irradiation? Could light induce disassembly?
It could be monitored by NMR.
4- Page 6, line 232-234, sentence “Due to… zeta potential”: it is not clear to me. Do the authors mean that the hemicyanines are more easily internalized in the C. albicans?
5- Page 8, line 298-300, sentence “Obviously,… HC10/CB[7]”: this statement is not clear to me. Do the authors have an explanation why?
6- Page 8, line 303-304 “which promoted… brighter fluorescence”: do the authors have a proof that the hemicyanines can reach the nucleus?
7- Fig. 5: the wavelength of irradiation should be mentioned.
Fig. 6: what is the ratio between the hemicyanines HC6 and CB[7]?
8- I think it is necessary to demonstrate the generation of ROS in the cells.
9- Fig. S1 & S2: the chemical shifts of the protons h & i are lower after complexation, which is in contrast to the chemical shifts of the protons a-e that moved to the high field. Is there any explanation why?
Are there traces of ethanol in some spectra (triplet and quadruplet at ca 1.25 & 3.75 ppm, respectively)?
10- Fig. S9 is not cited in the text. The data should be commented. What is the conclusion from these results?
Minor corrections:
- Typos: “antimicrobialantimicrobial” appears many time.
- It would be interesting to show the structure of CB[7].
- Fig. S3, S4, S5 and Table S1 should be cited in the text.
- Fig. S3: correct “Zeta potential” to Size.
- Fig. S4 & S5: correct “nm” to mV.
Round 2
Reviewer 1 Report
The revised version of the manuscript results in part improved with respect to the previously submitted one, but some of the requested changes and additions were not addressed.
For example, when I referred to the need of improving the language and the explanation for the Materials and Methods section, I mentioned a couple of sentences to change as examples but the indication was more general. There is an use of the verbs that makes the reading difficult. For example the sentence at line 76/77 “Hemicyanine HC6 or HC10 were configured with DMSO to a stock liquid with a solubility of 10 mM” what exactly does it mean? That stock solution of HC were prepared in DMSO at 10 mM concentration? If yes, please rephrase it.
Also sentence at line 80 is not completely clear: does “Then added 5 μL of hemicyanine HC6 or HC10” mean “5 μL of hemicyanine HC6 or HC10 mean were added”? If yes, please rephrase.
It lacks appropriate language also the sentence at line 236-238: “When HCs/CB[7] complex bound to pathogenic bacteria, the positive charges of hemicyanines were shielded by CB[7] to limiting the positive movement of microbes.”. What does it mean “to limiting the positive movement of microbe”? Please, rephrase correctly
Moreover, the authors reply to the questions relative to the NMR characterization of the complexes is not completely satisfying. They actually made the correction on the direction of the signal shifts upon complexation, as requested, but they do not explain how these shifts can justify the geometry of the HC/CB complexes and the positions of the CB around the HC molecule. They simply replied to me that based on the literature, the changes they observe by NMR together with the data collected with other techniques demonstrate the complexation. I have no doubt about the occurring complexation of HC by CB, but the authors represent in the figure a well-defined structure of the complex with the CB molecules precisely disposed along the HC molecules. This must be justified on the basis of the data and in principle, the inclusion of the aromatic units into the CB cavity should cause an upfield shift of the aromatic protons of HC, while a downfield shift was observed. Is there an explanation for this?
Also my request of explicitly describing the aggregation of HC molecules when the measurements by DLS are reported seems to have not been addressed. However, I continue to think that, also for the impact of this behaviour of the HC molecules on their spectroscopic properties, the fact that these molecules are in the form of aggregates in aqueous solutions must be explicitly mentioned not limiting the point to the simple number relative to the diameter estimated by DLS.
The new sentence added to the text by authors at line 252-256 seems to contain an important error, if I could correctly follow their reasoning. They state “…Whereas upon light irradiation, the colonies on the plates gradually decreased with the increase of CB[7] concentration in the HC6/CB[7]”, but I think that the colonies increase when CB increases because, by complexing HC6, it reduces its ability in producing ROS and then its antimicrobial activity.
Another request in my previous comments was not addressed. The authors simply replied to me in the cover letter but did not improve the text in the part relative to the investigation of interaction between HC/CB complexes and E. Coli (lines 290-306). They wrote in the letter “When we observed the interaction between HC/CB[7] with microorganisms, the unstained HC/CB[7] would be washed. ...” If this point is not explicitly explained in the manuscript, the description reported in this paragraph remains counterintuitive. It is not necessarily expected that the reader imagines this “washing step” if not reported. The authors can not simply limit to answer to me, they must make clear the text to all.
In my opinion, when all these points will be exhaustively addressed, the manuscript will be in the form suitable for publication on Nanomaterials.
Reviewer 2 Report
The authors responded to my questions and comments - I believe the manuscript is much improved.
Author Response
Thank you for your positive comments.
Reviewer 3 Report
In the revised version of the manuscript entitled "Regulation of Antimicrobial Effect of Hemicyanine-based photosensitizer via Supramolecular Assembly" the authors have addressed the reviewer comments. I suggest publication of this work in Nanomaterials. Beforehand, I would like the authors to show in Fig. S9 the damage on E. coli. It is not obvious to me.
